# Antihyperglycemic and Lipid Profile Effects of *Salvia amarissima* Ortega on Streptozocin-Induced Type 2 Diabetic Mice

**DOI:** 10.3390/molecules26040947

**Published:** 2021-02-11

**Authors:** Jesus Ivan Solares-Pascasio, Guillermo Ceballos, Fernando Calzada, Elizabeth Barbosa, Claudia Velazquez

**Affiliations:** 1Unidad de Investigación Médica en Farmacología, UMAE Hospital de Especialidades-2° Piso CORSE Centro Médico Nacional Siglo XXI, IMSS, Av. Cuauhtémoc 330, Col. Doctores, CP 06720 Ciudad de México, Mexico; jeivsp@gmail.com; 2Instituto Politécnico Nacional, Escuela Superior de Medicina, IPN, Salvador Díaz Mirón esq. Plan de San Luis S/N, Miguel Hidalgo, Casco de Santo Tomas, CP 11340 Ciudad de México, Mexico; gceballos@ipn.mx (G.C.); rbarbosa@ipn.mx (E.B.); 3Área Académica de Farmacia, Instituto de Ciencias de la Salud, Universidad Autónoma del Estado de Hidalgo, Km 4.5, Carretera Pachuca-Tulancingo, Unidad Universitaria, CP 42076 Pachuca, Mexico; claudiav@uaeh.edu.mx

**Keywords:** *Salvia amarissima* Ortega, diabetes mellitus, antihyperglycemic activity, lipid profile

## Abstract

*Salvia amarissima* Ortega was evaluated to determinate its antihyperglycemic and lipid profile properties. Petroleum ether extract of fresh aerial parts of *S. amarissima* (PEfAPSa) and a secondary fraction (F6Sa) were evaluated to determine their antihyperglycemic activity in streptozo-cin-induced diabetic (STID) mice, in oral tolerance tests of sucrose, starch, and glucose (OSTT, OStTT, and OGTT, respectively), in terms of glycated hemoglobin (HbA1c), triglycerides (TG), and high-density lipoprotein (HDL). In acute assays at doses of 50 mg/kg body weight (b.w.), PEfAPSa and F6Sa showed a reduction in hyperglycemia in STID mice, at the first and fifth hour after of treatment, respectively, and were comparable with acarbose. In the sub-chronic test, PEfAPSa and F6Sa showed a reduction of glycemia since the first week, and the effect was greater than that of the acarbose control group. In relation to HbA1c, the treatments prevented the increase in HbA1c. In the case of TG and HDL, PEfAPSa and F6Sa showed a reduction in TG and an HDL increase from the second week. OSTT and OStTT showed that PEfAPSa and F6Sa significantly lowered the postprandial peak at 1 h after loading but only in sucrose or starch such as acarbose. The results suggest that *S. amarissima* activity may be mediated by the inhibition of disaccharide hydrolysis, which may be associated with an α-glucosidase inhibitory effect.

## 1. Introduction

Diabetes mellitus (DM) type 2 is a complex chronic metabolic disease, characterized by hyperglycemia, and produced by a deficiency or loss of insulin activity; DM is generally accompanied by alterations like dyslipidemia, hypertension, obesity, or hyperinsulinemia [1,2,3]. According to estimates of the International Diabetes Federation (IDF), in 2019, more than 463 million people had DM (9.3% of the global population) [1,2]. In terms of DM cases, Mexico is the sixth highest ranked country in the world (12.8 million case), and DM is the second highest cause of mortality [2,3]. Treatment of DM includes adoption of healthy habits and is accompanied by pharmacological therapy [2,4], including biguanides, sulfonylureas, glitazones, α-glucosidase inhibitors, dipeptidyl peptidase-IV (DPP-IV) inhibitors, and incretin analogue therapies. The search for novel effective antidiabetic agents that avoid or reduce the side effects of these compounds is a continuous undertaking [5,6].

The use of traditional medicine, based on medicinal plants or phytotherapy, particularly in developing countries, includes a large number of individual medicines and, in some cases, substitute allopathic treatments, because the latter are less economically accessible than medicinal plants [7]. In Mexico, it has been reported that approximately 383 plant species are used for treatment of DM; however, not all of these have been systematically studied to assess their clinical or preclinical efficacy [8].

Due to lack of information of medicinal plants, it is important to provide more information about Mexican flora and their use in traditional medicine due to their health implications, including their toxicity, and to establish if they are safe or effective. The information obtained in this study could be used to expand the knowledge and use of *Salvia amarissima* Ortega. Recent studies of potential antidiabetic plants were related to diverse mechanisms of how they generate antihyperglycemic activity, such as α-glucosidase inhibitors, protein tyrosine phosphatase 1B inhibitors (PTP-1B), insulin secretion, or inhibition of DPP-IV [9,10,11,12,13,14,15,16,17,18,19,20]. Rural populations use herbal products or combine them with allopathic treatments for the control of blood glucose levels [21,22]. These treatments include species such as *Salvia amarissima* Ortega (Syn. *Salvia circinata* Cav.), which is a perennial aromatic shrub native to Mexico, commonly known as “bretónica”, “diabetina”, “prodigiosa”, “salvia”, or “insulina”, and used in Mexican traditional medicine for the treatment of ulcers, helminthiasis, and diabetes [9,21,23,24]. In previous studies, medium and high polarity extracts were isolated from *S. amarissima*: phenolic compounds (rosmarinic acid, rutin, pedalitin, apigenin-7-*O*-β-d-glucoside, 6-hydroxyluteoline, 2-(3-4-dimethoxyphenyl)-5-6-dihydroxy-7-methoxy-4*H*-chromen-4-one, 5,6,4′-trihydroxy-7-3′-dimethoxyflavone, and isoquercitrin) and terpenoids (amarisolides A–G, amarissinines A–C, acetylamarisine B, teotihuacanin, α-bourbanene, β-caryophyllene, δ-elemene, α-caryophyllene, β-selinene, germacrene D, and spathulenol [14,15,24,25,26]. Among these, amarisolide A, rutin, pedalitin, apigenin-7-*O*-β-d-glucoside, and 2-(3-4-dimethoxyphenyl)-5-6-dihydroxy-7-methoxy-4*H*-chromen-4-one are α-glucosidase inhibitors. In addition, amarisolide A, rutin, rosmarinic acid, isoquercitrin, and pedalitin are protein–tyrosine phosphatase 1B inhibitors (PTP-1B) [14,24]. It is important to highlight that low polarity extracts of fresh aerial parts have not been studied; however, the study of non-polar compounds is scarce, including when they showed effects in pre-clinical sub-chronic studies [9,15,24]. The focused polarity of this study provides information for the selection of the best conditions for potential herbal preparations or pharmaceutical products obtained from *S. amarissima*.

The aim of this study was to evaluate the antihyperglycemic and lipid profile effects of the non-polar extract of *Salvia amarissima* and its active fraction in male normoglycemic and streptozocin-induced diabetes mellitus type 2 mice.

## 2. Materials and Methods

### 2.1. General Information

Methanol (CC: 904903), ethyl acetate (CC:10382681), and petroleum ether (CC: 8032324) solvents were purchased from J.T. Baker^TM^ (Thermo Fisher Scientific, Waltham, MA, USA). Streptozocin (75% α-anomer basis, PN: S0130-5 G), nicotinamide (99.5%, PN: 47865-U), glucose (anhydrous, PN: D9434-1 Kg), sucrose (99.5% GC, PN: S9378-1 Kg), starch (CC 336155-1 Kg), acarbose (PN: PHR1253-500 mg), and canagliflozin (95%, PN:721174-1 G) were obtained from Sigma-Aldrich^®^ (Saint Louis, MO, USA). Silica gel (high purity grade (7734), pore size 60 Å, 70,230 mesh; CC: 391484-5 Kg) was purchased from Merck^®^ (Merck^®^, Darmstadt, Germany).

### 2.2. Plant Material

*Salvia amarissima* Ortega fresh aerial parts were collected in San Gregorio Atlapulco, Xochimilco, Mexico. The plant was identified by M.Sc. Santiago Xolalpa from the herbarium of the National Medical Center XXI Century, Mexican Institute of Social Security (IMSS) in México City, and a voucher specimen (16263) was made and deposited at the herbarium.

### 2.3. Obtaining the Petroleum Ether Extract of Salvia amarissima

Fresh aerial parts of the plant were cut (3 kg) in small pieces and then all were added to 9 L of petroleum ether to perform a metabolite extraction by thorough maceration at room temperature for one week; this extraction process was performed twice [10]. At the completion, the extract was collected through filtration and concentrated at 40 °C in a rotary evaporator coupled to a vacuum system (Büchi Labortechnik AG, Flawil, Switzerland); the extract was named PEfAPSa (petroleum ether extract of fresh aerial parts of *S. amarissima*). The final dry weight of the extract was 8.01 g of yellow wax with a yield of 0.27%.

### 2.4. Column-Chromatography and Phytochemical Screening

To better understand the activity of PEfAPSa, the dry extract was subjected to column chromatography using ethyl acetate (EtOAc) and methanol (MeOH) solvent elution over silica gel 60 G, successively using 150 mL of petroleum ether-EtOAc (100:0 to 0:100) for each fraction, followed by a mixture of EtOAc–MeOH (90:10 to 50:50). As result, seven fractions were obtained, namely F1Sa, F2Sa, F3Sa, F4Sa, F5Sa, F6Sa, and F7Sa. Of these, based on the results of the acute assays of the fractions obtained, only Fraction 6 (F6Sa) was purified by preparative thin layer chromatography (TLC) silica gel 60F-254 (Merck), using a petroleum ether–acetone (85:15) mixture as the mobile phase to obtain the secondary metabolites. The process was repeated as needed with the remaining extract to obtain the largest number of fractions for in vivo experiments and preliminary analysis of pure compounds. Preliminary qualitative identification of pure compounds was conducted in terpene compounds and was performed by TLC using sulfuric acid (10% in water) and/or ceric sulphate as revelators; pure compounds and terpenes standard had an Rf of 0.14 and were positive to ceric sulphate, indicating the presence of terpene compounds in F6Sa, but they were not yet characterized [27].

### 2.5. Animal In Vivo Assays

Male (for acute and sub-chronic assays, and oral tolerance tests (OTTs)) and female (for toxicity) Balb/c mice (20–25 g) were obtained from the animal bioterium of the National Medical Center XXI Century, Mexican Institute of Social Security (IMSS) in Mexico City, where animals were raised and maintained under standard laboratory conditions (conditions (12 h/12 h light/dark cycles; temperature, 25 ± 2 °C; humidity 55–80%) according to the Mexican Official Norm [10,28]. Food and water were provided ad libitum. The experimental protocol was registered and approved by the Local Research Committee with institutional register number FIS/IMSS/PROT/PRIO/19/110.

### 2.6. Toxicity Test

To determine if the PEfAPSa could be used safely, the acute toxic class method test 423 from the OECD [29] was performed. Fifteen Balb/c mice (only females), fasted overnight but allowed free access to water, were randomly assigned to the following groups of three mice each (*n* = 3): (I) negative control only administered with distilled water, and also healthy mice administered with extract in a single dose at (II) 3 mg/kg b.w., (III) 30 mg/kg b.w., (IV) 300 mg/kg b.w., and (V) 3000 mg/kg b.w. Mice were not fed for 4 h after administration. Signs of toxic effects and/or lethality were observed in the first 6 h after single administration and then for 48 h, and mice were examined for neurotoxic signs such as dizziness, lethargy, or aggressiveness. At the 0, 2, and 4 h mark after the load of the extract, glycemia was measured in all the treatment groups. The general health conditions of mice were observed daily for a period of 14 days for mortality, toxic effects, and/or changes in behavioral patterns. At the end of the experiment, the animals were sacrificed in a CO_2_ saturation chamber. Then, internal major organs (stomach, gut, kidney, spleen, and liver) were extracted, macroscopical observations were performed, and the relative organ weight was calculated [30,31] using the following formula:$$\mathrm{Relative}\;\mathrm{organ}\;\mathrm{weight}=\frac{\mathrm{Organ}\;\mathrm{weight}\;\left(g\right)}{\mathrm{Body}\;\mathrm{weight}\;\mathrm{of}\;\mathrm{the}\;\mathrm{animal}\;\mathrm{on}\;\mathrm{sacrifice}\;\mathrm{day}\;\left(g\right)}\;\times 100$$

### 2.7. Nicotinamide–Streptozotocin-Induced Experimental Hyperglycemia in Mice

Experimental hyperglycemia was induced in male Balb/c mice intraperitoneally by injection of streptozotocin (STZ) at a dose of 100 mg/kg b.w. dissolved in citrate buffer solution (pH 4), which was maintained in ice prior to use. Fifteen minutes after the nicotinamide (NA) injection, intraperitoneal injection of nicotinamide at a dose of 240 mg/kg b.w. dissolved in distilled water were made in each male mouse at day zero. One week later, blood samples were taken from each animal, and those mice that showed blood glucose ≥210 mg/dL were considered as hyperglycemic and with experimental diabetes and were selected for the following studies [27].

### 2.8. Acute Antihyperglycemic Assay

To determine how PEfAPSa and F6Sa generate a diminution of glycemia, acute assays in diabetic animals were performed. Animals were divided into experimental groups of six animals each (*n* = 6), and treatments were administrated by the intragastric route with a cannula in a single administration divided into different groups of the following treatments: acarbose, PEfAPSa, and fractions 1–7 of *S. amarissima*. The quantities applied to the animal groups are listed below.

Acarbose acute activity: (I) streptozocin-induced diabetic (STID) mice control, (II) STID mice + acarbose 50 mg/kg b.w., and (III) STID mice + acarbose 100 mg/kg b.w. This was performed at the same time as the PEfAPSa assay.

Acute activity of PEfAPSa: (I) STID mice control, (II) STID mice + acarbose 100 mg/kg b.w. as drug control group, (III) STID mice + PEfAPSa 50 mg/kg b.w., (IV) STID mice + PEfAPSa 100 mg/kg b.w., and (V) STID mice + PEfAPSa 300 mg/kg b.w. This was performed at the same time as the acarbose assay.

Acute assay of fractions: (I) STID mice control, (II) STID mice + acarbose 100 mg/kg b.w., (III) STID mice + F1Sa 50 mg/kg b.w., (IV) STID mice + F2Sa 50 mg/kg b.w., (V) STID mice + F3Sa 50 mg/kg b.w., (VI) STID mice + F4Sa 50 mg/kg b.w., (VII) STID mice + F5Sa 50 mg/kg b.w., (VIII) STID mice + F6Sa 50mg/kg b.w., and (IX) STID mice + F7Sa 50mg/kg b.w.

Acute activity of F6Sa: (I) STID mice control, (II) STID mice + acarbose 100 mg/kg b.w., (III) STID mice + F6Sa 25 mg/kg b.w., (IV) STID mice + F6Sa 50 mg/kg b.w., (V) STID mice + F6Sa 100 mg/kg b.w.

The healthy mice group was administrated with vehicle (Tween 80:H_2_O, 2:98). Blood samples were collected at 0, 1, 3, 5, and 7 h after the first administration of each treatment. These assays were performed at the same time [10,27].

### 2.9. Sub-Chronic Assay

The sub-chronic assay was performed on diabetic mice to observe the behavior of the treatments, contrasting healthy and diabetic animals. Diabetic mice were divided into 4 experimental groups of six animals each (*n* = 6), and treatments were administrated daily by the intragastric route for six weeks as follows: (I) diabetic mice administered only with vehicle (Tween 80:H_2_O, 2:98), (II) PEfAPSa (50 mg/kg b.w.), (III) F6Sa (50 mg/kg b.w.), (IV) acarbose (100 mg/kg b.w.) as reference group, and (V) healthy group administered only with vehicle. Blood glucose levels (mg/dL) and body weight variations (g) were evaluated every 7 days, and every 15 days, evaluations were made of glycated hemoglobin (HbA1c), triglycerides (TG), and high-density lipoprotein (HDL). HbA1c, TG, and HDL blood samples were obtained by puncture in the caudal vein to obtain blood samples. HbA1c was measured by Clover A1c Analyzer (Infopia Co., Ltd., Anyang, Korea) using 40 μL, and TG and HDL were measured with the lipid panel of Cardiocheck Professional PA Silver (Polymer Technology Systems, Whitestown, IN, USA) using 40 μL of blood. At the end of the experiment, the animals were sacrificed in a CO_2_ saturation chamber, organs were macroscopically evaluated [9,28], and the relative organ weight of each animal was calculated with the equation in Section 2.6 [30,31].

### 2.10. Oral Tolerance Test (OTT)

To identify part of the probable mechanism by which the plant exerts its activity, inhibition of α-glucosidases (with sucrose and starch) or the glucose transport mechanism (glucose), studies were conducted consistent with other plants species from diverse families, such as *Calea ternifolia* [9], *Annona cherimolla* [10], or *Scutellaria baicalensis* [11], for example, that had been studied as antihyperglycemic agents [8,9,10,11,12,13]. The OTT technique was performed after an overnight fasting (18 h) of normoglycemic mice. Mice were divided into groups of six animals each and administrated with the following treatments: (I) Tween 80:H_2_O (2:98), (II) PEfAPSa (50 mg/kg b.w.), (III) PEfAPSa (100 mg/kg b.w.), (IV) F6Sa (50 mg/kg b.w.), (V) F6Sa (100 mg/kg b.w.), (VI) acarbose (100 mg/kg b.w.) as control for oral sucrose tolerance test (OSTT) and oral starch tolerance test (OStTT), or (VII) canagliflozin (100 mg/kg) control for oral glucose tolerance test (OGTT). Baseline (time 0 h) was set before the administration of the treatment previously described; after 30 min, the respective treatment was administered to the mice as follows: glucose (1.5 g/kg b.w.), sucrose (3 g/kg b.w.), and starch (3 g/kg b.w.). Blood samples were collected at 1, 2, and 4 h after the glucose load [9,27].

### 2.11. Collection of Blood Glucose and Determination of Blood Glucose

Blood samples for each assay were collected from the caudal vein by a small incision at the end of the tail, and the glucose levels (mg/dL) were measured using a commercial glucometer (Accu-chek Performa, Roche Diagnostics GmbH, Manheim, Germany).

### 2.12. Statistical Analysis

Data are expressed as the mean (±) with the standard error of the mean (SEM). Statistical significance was determined by one-way analysis of variance (ANOVA) followed by the Dunnet test (GraphPad Prism Version 6.01; GraphPad Software Inc., La Jolla, CA, USA), with post hoc test * *p*-values < 0.05 indicating significant differences between experimental group means.

## 3. Results

### 3.1. In Vivo Assays

#### 3.1.1. Acute Oral Toxicity Test of the PEfAPSa

The single administration of PEfAPSa at all tested dose in female mice by gavage did not induce lethality or behavior alterations, macroscopic tissue injury, or total weight loss (data not shown) during the 14 day observation period; therefore, the LD_50_ value of PEfAPSa was established as >3000 mg/kg b.w. in female mice when administered by the oral route according to OECD 423 guidelines [28]. For our study, the parameter of importance was the glycemia of healthy mice at the doses established in the OECD guidelines. Table 1 shows the results obtained in which there was no hypoglycemia, that is, the principal secondary effect of the hypoglycemic drugs. At doses of 300 and 3000 mg/kg b.w., at the second hour of treatment an increase in glycemia was observed, before a return to the initial values. Because of the prevention of mortality, the therapeutic window and range of use were wide.

#### 3.1.2. Evaluation of Acute Antihyperglycemic Effect of PEfAPSa

The acute antihyperglycemic activity was evaluated to determine the potential antihyperglycemic effect of PEfAPSa and to select the dose for the subsequent sub-chronic assay, which indicates the behavior of the PEfAPSa at different doses (Figure 1A). Furthermore, this assay was used to evaluate two doses of acarbose to determine that to be used as the drug control. The extract was evaluated at three single doses of 50, 100, and 300 mg/kg b.w. Among these, two doses, namely 50 and 300 mg/kg b.w., generated a significant decrease in hyperglycemia in STID mice in all of the measured times. The effect was comparable to that obtained with acarbose at a dose of 100 mg/kg b.w. For subsequent studies, the dose of 50 mg/kg b.w. of PEfAPSa was selected, considering it is a lower dose with major and constant effects with time. In contrast, those diabetic mice that were administered with PEfAPSa at a dose of 100 mg/kg b.w. showed increased levels of blood glucose compared even to those results obtained from the group of diabetic mice without treatment (375 mg/dL), for which results were discarded. The dose selected for acarbose was 100 mg/kg b.w. because it had a similar effect as that observed with PEfAPSa at a dose of 50 mg/kg b.w. until the last hour of evaluation. The dose of 300 mg/kg b.w. did not generate hypoglycemia as seen in the toxicity assay in healthy mice, as shown in Figure 1A.

#### 3.1.3. Evaluation of Acute Antihyperglycemic Activity of the Secondary Fractions (F2Sa, F5Sa, and F6Sa) of *Salvia amarissima*

Once the effect of PEfAPSa and acarbose were evaluated and adequate doses were selected, the extract was subjected to column chromatography, as described in Section 2, obtaining seven fractions named F1Sa, F2Sa, F3Sa, F4Sa, F5Sa, F6Sa, and F7Sa. These were evaluated to identify those that had antihyperglycemic activities similar to that of PEfAPSa. Figure 2 shows the fractions that had some antihyperglycemic activity. The fractions that showed activity were F2Sa, F5Sa, and F6Sa. F2Sa and F5Sa had activity at the first and third hours, respectively, but lost their activity after that time. F6Sa showed a reduction of blood glucose from the third hour and maintained its activity, similar to the case of PEfAPSa, and in the last two hours to the case of the acarbose control group. This fraction (F6Sa) was selected for later assays (Figure 1B).

#### 3.1.4. Acute Activity of F6Sa of *Salvia amarissima*

After demonstrating the activity of F6Sa, it was evaluated at three different doses (25, 50, and 100 mg/kg b.w.), to identify which of these had better antihyperglycemic activity, and the behavior of the fraction in STID mice. The single dose of F6Sa at 50 mg/kg b.w. (Figure 1C) induced a decrease in glycemia to almost 324 mg/dL, 5 h after administration, compared with the mice vehicle group (386 mg/dL), showing similar activity at the fifth hour to the acarbose control group (323 mg/dL). Diabetic animals treated with F6Sa at doses of 25 and 100 mg/kg b.w. increased glycemic levels to 449 mg/dL and 428 mg/dL, respectively, compared to the vehicle group. Because of these results, the selected dose was 50 mg/kg b.w. for later assays. As seen in Figure 1C, F6Sa showed similar behavior to the case in which there was no dose-dependent relationship between the evaluated doses.

### 3.2. Subchronic Assay

#### 3.2.1. Organ Weight Measurement of PEfAPSa, F6Sa, and Acarbose on STID Mice of Subchronic Assay

As seen in Table 2, the analysis of the organs with the sub-chronic assay indicate that, with the exception of the spleen, differences exist between the diabetic control group and the diverse treatments. In the case of the liver in PEfAPSa and F6Sa, a diminution of the organ weight occurred, but this was similar to the healthy mice control, and in the macroscopic analysis there were no changes in the case of the STID mice liver, which had a pale color. PEfAPSa generated a reduction of the kidneys, but there was no relationship with the results shown in Figure 1C. Finally, for the results of the intestine, a significant increase in weight occurred in the diabetic control. This could be explained as a result of the polyphagia inherent with diabetes mellitus. PEfAPSa and F6Sa maintained similar values to that of the healthy control group. Finally, PEfAPSa and F6Sa generated a diminution of the intestine weight compared to the diabetic control group and the acarbose group and was also similar to the healthy group (Table 2).

#### 3.2.2. Subchronic Assessment of Blood Glucose with PEfAPSa, F6Sa, and Acarbose on STID Mice

Once the active doses of the diverse treatments of the acute assays were selected, the sub-chronic assay was performed. This test validated the STID mice model to measure different parameters that are altered in diabetes mellitus, such as glucose, HbA1c, TG, HDL, and organ alterations. In the case of validation of the diabetic model, the STID mice of the control group did not undergo a recovery, and glucose levels suggested that the development of experimental diabetes in animals was constant over time, with glycemia increasing from 347 to 471 mg/dL at the end of the experiment (Figure 2A). This is similar to poorly controlled diabetic patients. In contrast, the healthy vehicle group did not show an increase in blood glucose. The acarbose group showed constant glycemic levels (329 to 313 mg/dL) in all of the assays. The PEfAPSa group showed major activity compared to the acarbose group, with a decrease from 323 to 262 mg/dL. The F6Sa group showed major activity from the first week of treatment (259 mg/dL), maintaining glucose levels significantly lower than those of the control group at the end of the experimental period, and showed similar activity to the acarbose group (313 mg/dL). PEfAPSa and F6Sa showed better activity compared to the acarbose control group in the six weeks of treatment (Figure 2A).

#### 3.2.3. Glycated Hemoglobin Measurement of PEfAPSa, F6Sa, and Acarbose on STID Mice of Subchronic Assay

The animals of the sub-chronic assay were also used for complementary analysis. HbA1c was used in the complementary test to confirm and predict the degree of progression of the disease (Figure 2B). The evaluation of the HbA1c for the sub-chronic assay indicated a partial relationship with the glucose level evaluation of previous results (Figure 2A). The healthy group maintained values from 4.5 to 5.3%, whereas the diabetic group showed values of 7.6 to 9%, and the acarbose, PEfAPSa, and F6Sa groups generated similar values, of between 7.3 to 7.5%, as shown in Figure 2B. These results indicate that the evaluated treatments have similarities, as also seen in the results shown in Figure 2A,B.

#### 3.2.4. Measurement of Lipids Parameters in Subchronic Assay of PEfAPSa, F6Sa, and Acarbose on STID Mice

Other measured parameters that were found to have significant changes in diabetes mellitus were the TG and HDL. These parameters were also related in the metabolic syndrome and cardiovascular diseases [3]. The changes that generate *S. amarissima* in the lipid profile can serve to broaden the therapeutic uses of the plant. The variations observed in Figure 2C,D were measured in the sub-chronic assay.

Triglyceride Measurement of the Subchronic Assay of PEfAPSa, F6Sa, and Acarbose on STID Mice

The values obtained (Figure 2C) indicated that from the second week, all of the treatments showed a decrease in the values of TG. PEfAPSa and F6Sa maintained major activity for six weeks. They also maintained a lower concentration of this parameter and had a better effect than that observed with the acarbose group, which from the fourth week showed an increase in the TG values and at the end of the assay was similar to that of the STID mice group. This indicates that a reduction of the TG generates an improvement of diabetic management, decreasing the possibility of alterations in other parameters associated in a diabetic complex like dyslipidemias and/or cardiovascular complications, incorporating complementary activity for the use of the plant.

HDL Measurement of the Subchronic Assay of PEfAPSa, F6Sa, and Acarbose on STID Mice

The measured HDL values (Figure 2D) indicate that PEfAPSa and F6Sa, at the second week, increased the HDL values, followed by a similar decrease in the values for both cases. The combination of results could indicate an effective optimum control and reduction of cardiovascular disease associated with diabetes mellitus.

### 3.3. Oral Tolerance Test Assays

The principal mechanisms studied from the plants were focused on the inhibition of glucosidase enzymes and glucose transport mechanisms [7,8,9,10,11,12]. The α-glucosidase enzymes mostly used for the evaluations of the glucosidase enzymes were sucrose and starch, and glucose was used for the glucose transporter, because the enzymes are focused on the degradation of disaccharides and polysaccharides. These tests were carried out to evaluate the activity of the different glucosidase enzymes (Figure 3A,B) and of the glucose transport (Figure 3C).

#### 3.3.1. Oral Sucrose Tolerance Test (OSTT)

In the OSTT (Figure 3A), the principal source of glucose is from the degradation of the disaccharide sucrose; the healthy control group in the assay maintained a blood glucose average of 100 mg/dL, and the acarbose control group showed a decrease in glycemia to 81 mg/dL in the first hour and then to 65 mg/dL. The PEfAPSa did not induce a decrease in postprandial glycemia. Only F6Sa generated a diminution of the blood glucose levels that was similar to that of the healthy control group (121 mg/dL), as shown in Figure 3A. All of the treatments, with the exception of acarbose, returned to the values similar to those of the control group at the second hour. This indicates that part of the effect observed in acute and sub-chronic assays could be related to the inhibition of the hydrolysis of carbohydrates.

#### 3.3.2. Oral Starch Tolerance Test (OStTT)

In the case of the OStTT (Figure 3B), all of the treatments generated an inhibition of the glucose postprandial peak of the first hour after the starch loading, which was similar to the values of the acarbose group. The effects appeared to be more active at the doses of 50 mg/kg b.w. in both cases. The results suggest that if the carbohydrate is more complex, the activity of the treatment could be improved.

#### 3.3.3. Oral Glucose Tolerance Test (OGTT)

The results of the OGTT (Figure 3C) show that all of the evaluated treatments generated a reduction of the postprandial peak in comparison with the glucose group. Only PEfAPSa at 50 mg/dL generated an increase in blood glucose at the first hour with the dose of 50 mg/kg b.w. This indicated that the activity is not related to glucose transport.

## 4. Discussion

*S. amarissima* is a plant used in traditional Mexican medicine to treat DM. The background of the plant includes ethnomedical uses [21,23], and use in the evaluation of polar and medium polar extracts (MeOH:acetone and aqueose extracts), as well as in products obtained from these extracts (flavonoids and diterpens) [14,15,24,25,26]. The aforementioned studies focus on acute assays and some tolerances, while the present work provides information regarding acute, sub-chronic, and oral tolerance test assays of a non-polar extract and fraction. In addition, this study provides information regarding the behavior of HbA1c, TG, and HDL in a sub-chronic assay and information regarding the part of the mechanism through which they exert their effects.

The evaluation of the extract (PEfAPSa), according to acute toxic class method test 423 from the OECD guidelines [29], indicates that the estimated LD_50_ for the present study is determined as uncategorized because, at the maximum evaluated dose (3000 mg/kg b.w.), this dosage causes no mortality. The information provided by this assay is useful for choosing the range of doses that can be used for an acute assay in STID with non-polar petroleum ether extract. These results are in accordance with those of another research group [24], who described the LD_50_ of an aqueous extract (>5 g/kg). Additionally, in this experiment, the glycemia levels in the first 4 h of treatment indicated that no doses generated hypoglycemia, which is one of the side effects of the most common antidiabetic drugs [31].

The results of acute assays in STID mice, shown previously in Figure 1A–C, the model used in the assays conducted in this study, has been corroborated as a viable model for the evaluation of antidiabetic treatments where streptozotocin generates a disfunction in the pancreatic β-cells [10,30,32,33,34]. This work is focused on the evaluation of the non-polar petroleum ether extract (PEfAPSa) and its respective fractions, because studies focused on high-polarity compounds, such as flavonoids, or medium polarity compounds, such as diterpenes, possess antidiabetic activity, excluding the assessment of non-polar compounds [8,9,10]. In STID mice, single doses of petroleum ether extract (PEfAPSa) showed a diminution of glycemia at doses of 50 and 300 mg/dL, and a dose of 50 mg/dL was selected as the optimal dose because, according to the results, it seems that a dose–response relationship was not observed between them. Following analysis of PEfAPSa’s antihyperglycemic effects, it was fractionated in a chromatographic silica gel column, and seven fractions were obtained, named F1Sa–F7Sa.

From the evaluated fractions, the fraction that maintained a similar activity to that of the extract was fraction 6 (F6Sa) in STID mice acute assays. From the three different evaluated doses, the selected dose was 50 mg/kg b.w., which showed antihyperglycemic activity at the fifth and seventh hours of treatment, being similar in time to PEfAPSa. This similarity is related to the presence of compounds that generate a glycemia reduction in STID mice, and these results show that the non-polar compounds obtained from petroleum ether extract can be as effective as polar and non-polar extracts, while fresh parts can be used in place of dry parts [14,15,24,25,26]. The benefit of the extended time effect in acute assays of PEfAPSa and F6Sa indicates that these treatments could be administrated less frequently throughout the day, while still being effective. Additionally, as seen in the PEfAPSa acute assay, the results of F6Sa indicate that there was no a dose–response relationship, an effect described as hormesis because the response was independent of the dose or because there was a biphasic dose–response relationship. To definitively confirm the presence of hormesis, it is necessary to evaluate more doses to generate a complete dose–response curve [35,36]. After the identification of the active fraction (F6Sa), it was subjected to preparative thin-layer chromatography, from which some products were identified. Three of these were preliminarily identified as terpene compounds, which could be inferred by their positive reaction to ceric sulphate, in which the color observed changed from purple to red. Plants of this family (Lamiaceae) contain a high quantity of terpenoids (monoterpenes, diterpenes, and triterpenes) [37,38,39,40], as in the case of *S. amarissima*, having been identified as neo-clerodane diterpenes, such as amarisolides, amarissinins, and teotihuacanine [24,25,26,38,41].

The diabetes mellitus 2 experimental model was suitable because glycemia levels were maintained above 350 mg/dL [31,34], generating similar biochemical changes to diabetes in humans (glycemia and HbA1c) [31]. The evaluation of the effects in the sub-chronic assay, using the previously selected doses, indicated that PEfAPSa and F6Sa showed higher activity than the control (acarbose) during the time of the assay. Our evaluation of HbA1c suggests that PEfAPSa and F6Sa have similar effects to the acarbose control group, with the three groups showing similar assay results, as shown by the effects below. The similarity with acarbose indicates that products with activity over α-glucosidase enzymes could exist, therefore affecting the HbA1c levels [3].

DM2 is characterized by relative insulin deficiency and a decrease in lipoprotein due to a decrease in adiponectin, resulting in high levels of low-density lipoprotein and TG and low levels of HDL. As reported in Kidwai [3], a relationship exists between high levels of HbA1c and dyslipidemia; HbA1c increases the risk of hypertriglyceridemia by 2.69% [3] as well as the development of a high risk of cardiovascular disease. As seen with PEfAPSa and F6Sa, the reduction in glycemia in acute and sub-chronic assays, TG, and maintenance of HbA1c, suggest that these two therapies are viable treatments for diabetes mellitus and for the control of HbA1c values, avoiding its increase and for TG and HDL taking them closer to healthy mice values than diabetic animals.

The OTT was used to determine the treatments’ potential mechanisms of action. These assays are focused on various glucosidase enzymes for a specific kind of bond (α-glycosidic bond), and on glucose transport. As seen in previous results, the OGTT showed that the effects of the extract, fraction, and compounds are not related to the glucose transport because there were no changes in the glucose postprandial peak in the first hour. Furthermore, in the case of sucrose and starch, PEfAPSa and F6Sa inhibit the postprandial glucose peak, whereby the action of the treatments is focused on these enzymes (α 1–2 for sucrose and mixtures of α 1–4 and α 1–6 for starch), which is congruent with the previous results. These effects, in part, could be explained because the enzymes have significant permeability in the small intestine, which makes it possible for them to travel through the intestine or to another part of the body [42,43,44,45,46]. This explains how PEfAPSa and F6Sa can maintain the levels of hemoglobin in sub-chronic assays, reducing the levels of glucose obtained from the diet.

## 5. Conclusions

PEfAPSa and F6Sa showed activity in acute and sub-chronic assays of STID mice and in the regulation of the levels of blood glucose in the acute assay and HbA1c in the sub-chronic assay. Lipid parameters indicate that these could be effective complementary treatments for dyslipidemias associated with DM.

In the OTT, the activity is focused on sucrose and starch enzymes, and does not involve glucose transport. This indicates that the combination of secondary metabolites contained in the extract and the fraction are responsible for the antihyperglycemic activity in acute and long administration periods. Futures studies will evaluate different terpenic compounds and combinations of those products to find the best mix for a better treatment.

## Figures and Tables

**Figure 1 molecules-26-00947-f001:**
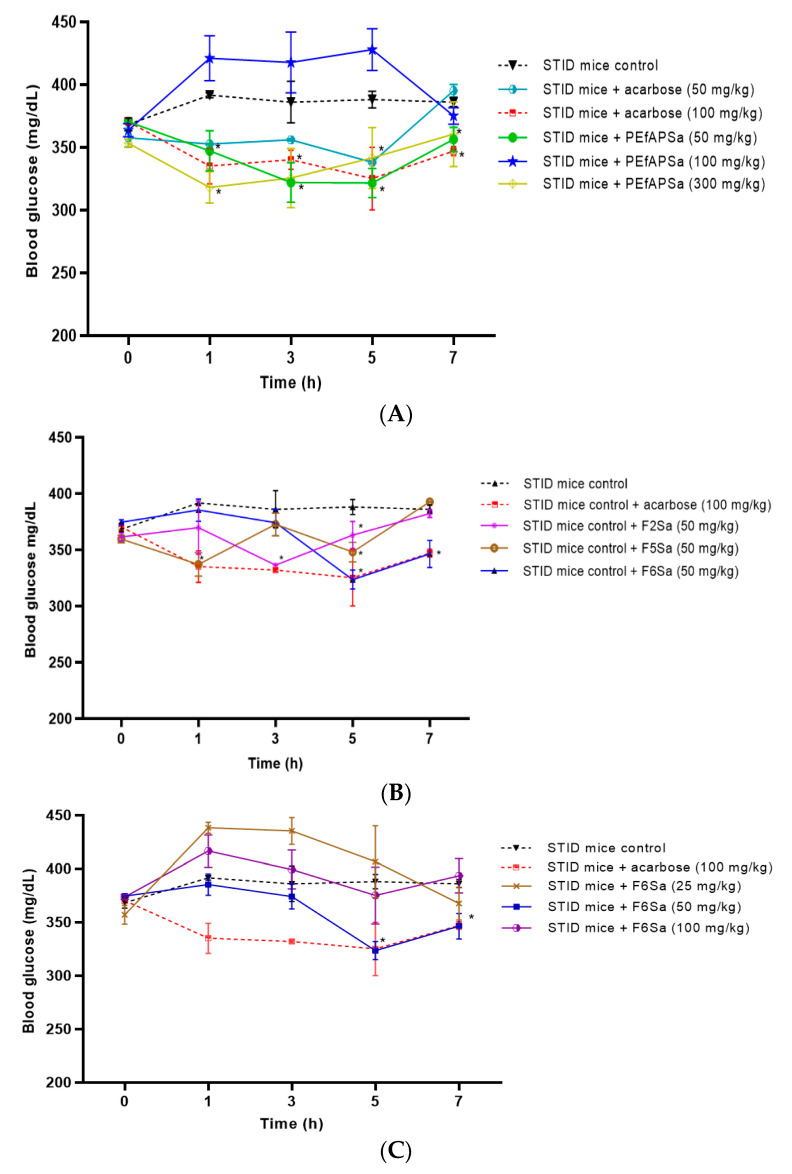
Acute assays of streptozocin-induced diabetic (STID) mice of (**A**) PEfAPSa in diabetic mice at doses of 50, 100, and 300 mg/kg b.w., and acarbose at doses of 50 and 100 mg/kg b.w.; (**B**) active fractions (F2Sa, F5Sa, and F6Sa) obtained from PEfAPSa, every fraction evaluated at the dose of 50 mg/kg; (**C**) effect of Fraction 6 (F6Sa) in diabetic mice at doses of 25, 50, and 100 mg/kg b.w. Each point represents the mean ± SEM, *n* = 6. * *p* < 0.05 represents significantly different two-way ANOVA followed by Dunnet’s post hoc test for comparison with respect to STID mice control group at the same time.

**Figure 2 molecules-26-00947-f002:**
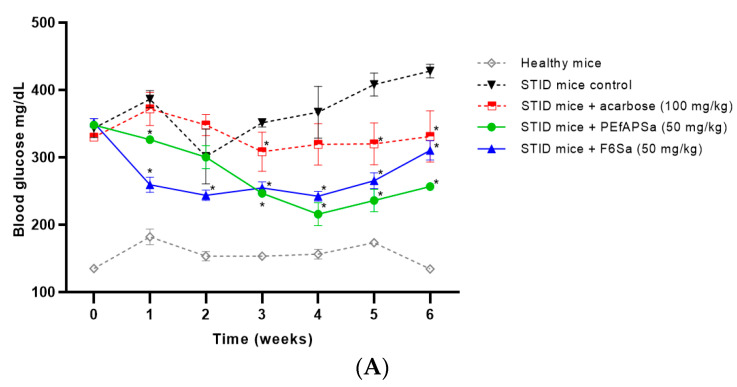
Subchronic assay results of the treatments: healthy mice, STID mice control group, STID mice + PEfAPSa 50 mg/kg b.w., STID mice + F6Sa 50 mg/kg b.w., and STID mice + acarbose 100 mg/kg b.w.; (**A**) glycemia in mg/dL, (**B**) HbA1c, (**C**) TG in mg/dL, and (**D**) HDL in mg/dL. Each point represents the mean ± SEM, *n* = 6. * *p* < 0.05 represents significantly different two-way ANOVA followed by Dunnet’s post hoc test for comparison with respect to STID mice control group at the same time and parameter measured.

**Figure 3 molecules-26-00947-f003:**
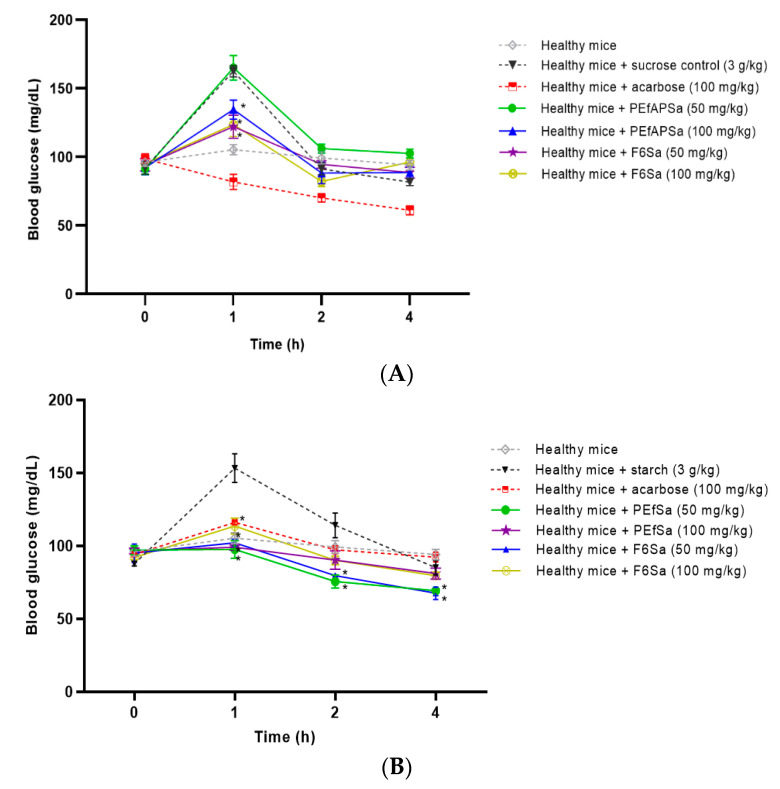
Oral tolerance test of healthy mice, PEfAPSa, at 50 and 100 mg/kg b.w., F6Sa at doses of 50 and 100 mg/kg b.w. in the subsequent carbohydrates. (**A**) Sucrose at a dose of 3 g/kg b.w., and acarbose at a dose of 100 mg/kg b.w. (**B**) Starch at a dose of 3 g/kg b.w. and acarbose at a dose of 100 mg/kg b.w. (**C**) Glucose at a dose of 1.5 g/kg b.w. and canagliflozin at a dose of 100 mg/kg b.w. Each point represents the mean ± SEM, *n* = 6; * *p* < 0.05 represents significantly different two-way ANOVA followed by Dunnet’s post hoc test for comparison with respect to healthy control group with carbohydrate load at the same time.

**Table 1 molecules-26-00947-t001:** Glycemia measurements in the toxicity assay of PEfAPSa.

Treatments					
Time (h)	Healthy Mice	Healthy Mice + PEfAPSa 3 mg/kg b.w.	Healthy Mice + PEfAPSa 30 mg/kg b.w.	Healthy Mice + PEfAPSa 300 mg/kg b.w.	Healthy Mice + PEfAPSa 3000 mg/kg b.w.
0	116.67 ± 3.96	119.33 ± 2.00	113.44 ± 2.37	108.78 ± 1.51	115.67 ± 2.00
2	114.78 ± 3.32	115.33 ± 2.62	117.33 ± 5.43	129.11 ± 3.19 *	151.56 ± 1.79 *
4	110.89 ± 1.63	109.67 ± 1.23	109.89 ± 1.15	112.89 ± 2.75	115.67 ± 1.62

Toxicity of the PEfAPSa at doses of 3, 30, 300, and 3000 mg/kg b.w. in healthy mice. * *p* < 0.05 represents significantly different one-way ANOVA followed by Dunnet’s post hoc test for comparison with respect to vehicle at the same time.

**Table 2 molecules-26-00947-t002:** Relative organ weight of the sub-chronic assay PEfAPSa.

Treatments					
Organs	Healthy Mice	STID + Vehicle	STID + PEfAPSa (50 mg/kg b.w.)	STID + F6Sa (50 mg/kg b.w.)	STID + Acarbose (100 mg/kg b.w.)
Pancreas	0.225 ± 0.054	0.121 ± 0.056	0.109 ± 0.035	0.111 ± 0.060	0.226 ± 0.100
Spleen	0.085 ± 0.016	0.115 ± 0.033	0.089 ± 0.005	0.096 ± 0.020	0.093 ± 0.032
Stomach	0.392 ± 0.073	0.227 ± 0.032	0.288 ± 0.338	0.136 ± 0.084	0.696 ± 0.217 *
Liver	1.073 ± 0.152	1.592 ± 0.183	1.220 ± 0.315 *	0.638 ± 0.587 *	1.465 ± 0.090
Kidneys	0.123 ± 0.022	0.430 ± 0.035	0.160 ± 0.014 *	0.170 ± 0.024 *	0.166 ± 0.012 *
Intestine	1.487 ± 0.051	3.390 ± 0.396	1.838 ± 0.842 *	0.824 ± 1.445 *	2.442 ± 0.450 *

Relative organ weight of the sub-chronic assay PEfAPSa, F6Sa, and acarbose on STID mice. Subchronic activity of PEfAPSa 50 mg/kg b.w., F6Sa 50 mg/kg b.w., and acarbose 100 mg/kg b.w. in STID mice. Each point represents the mean ± SEM, *n* = 6. * *p* < 0.05 represents significantly different one-way ANOVA followed by Dunnet’s post hoc test for comparison with respect to STID mice control of the respective organ. Relative organ weight was calculated as (organ weight (g)/body weight of animal on sacrifice day (g)) × 100.

## Data Availability

The data presented in this study are available on request from the corresponding author.

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
