# Peer review of "Antihyperglycemic and Lipid Profile Effects of Salvia amarissima Ortega on Streptozocin-Induced Type 2 Diabetic Mice"

_molecules, 2021, doi:10.3390/molecules26040947_

Round 1
Reviewer 1 Report
The article submitted for review (Molecules-1055204) is a valuable contribution to the search for natural sources of substances at hypoglycemic and lipid profile properties, to prevent and alleviate the symptoms of diabetes. The topic taken is important and current.
The authors made a toxicological assessment of the non-polar extract of Salvia amarissima and its active fraction as well as their hypoglycemic effect in diabetic mice. However, the manuscript requires adjustments.
The main comments to the reviewed article are:
- The lack of characteristics of the qualitative and quantitative composition of the studied petroleum ether extract of fresh aerial parts of Salvia amarissima Ortega (PEfAPSa) and a secondary fraction (F6Sa).
- The sequence of presentation of particular sections is improper. The Materials and methods section should be presented before the description of the results, as it will allow for a better understanding of the results obtained, thanks to the knowledge of the research models used.
- It should be clearly marked that the applied doses were expressed as mg per kg body weight (mg/kg b.w.)in the whole text.
The minor remarks to the reviewed manuscript are as follows:
- Page 2 line 50: The full name of the DPPIV abbreviation used is missing, as well as TLC (page 11, line 301).
- Page 6, figure 6: The results of the statistical analysis in this graph are not shown. Instead of EPfAPS should be PEfAPS.
- Page 10, points 4.3 and 4.4 seem to be the same with the exception of the vacuum system used. Point 4.3 seems to unfinished.
- Page 11 line 306: The wrong reference was used.
- Page 11, line 321: Instead of the word "sing" should be "sign"
- Page 11, line 316-317: Unclear sentence, did you mean the Acute Toxic Class Method (test 423)?
Author Response
"Please see the attachment"

Reviewer 2 Report
Although the study of Pascasio et al., has the potential to be of interest, at this technical level and the level of the leading idea elaboration, it is not acceptable for publication. The authors should rewrite and reorganised the manuscript in order to make it clearer and more representable. Additionally, extensive editing of English language and style is required, and please recheck the spelling. It would be much better if the figures are organised in panels, in that case it would be easy to follow.
Author Response
"Please see the attachment"

Reviewer 3 Report
Jesus Iván Solares Pascasio and colleagues studied the antihyperglycemic effects of Salvia amarissima Ortega extract in an experimental diabetes mellitus type 2 model. Although the research study might be promising some major issues need to be addressed. Methods should be more clearly described, some results are missing and some experiments are not described in the material and method section, etc. Specific comments are given below:
- Abbreviations should be defined when first mentioned, e.g. TLC.
- In line 74 please correct the typo DL50
- From the materials and methods section it is evident that authors have performed the toxicity test, results of which are not presented.
- Page 3, on figure 2 results of acute antihyperglycemic activity for fractions F2Sa and F5Sa are shown, however those experiments are not described in the material and methods section. Please correct.
- Based on the presented data it seems that lowest and highest F6Sa dose increased glycemic level while middle dose decreased it. This should be discussed.
- Page 6, Figure 6 – It is recommended to calculate and present on the graph the relative organ weight, i.e. organ to body weight ratio, as this is optimum for most of the organs for prediction of toxicity.
- It seems that lines 283-289 are duplication of lines 290-296.
- On page 11 authors specify that only F6Sa fraction was purified, however the rationale of selecting this fraction over the other 6 fractions is not clear.
- Page 11, line 308 please correct “OSST” to “OSTT”
- In line 316 please remove repeated words “acute oral”
- Assessment of OSTT, OStTT and OGTT should be described in the materials and methods section.
- Moreover, the author should try to characterize the active compound(s) in the F6Sa fraction.
- Finally, English should be improved.
Author Response
"Please see the attachment"

Round 2
Reviewer 2 Report
Pascasio et al. improved the manuscript after first round of revision, but not enough to be acceptable for publication. Although the idea and the concept of the paper is good, the manuscript is lacking in the organisation, presentation and interpretation of the results. I hope that the list of suggestions will help the authors to improve the paper for the publication in another journal.
Section 2. Materials and methods
The description of methodology used for determinations of glycosylated hemoglobin, triglycerides, high-density lipoprotein (HDL) and urine glucose levels should be incorporated in the manuscript.
Line 164: Diabetic mice were divided into 4 experimental groups, not in 7 like you stated
Line 169: data for the urine glucose levels aren’t presented (while you are mentioned it in the Discussion section line 482), please incorporate this in the manuscript as well as the methodology used for this analysis.
Section 2.10. Oral tolerance test (OTT) the description of the experiment doesn’t match with presented results in the Figure 3A-C. In the section 2.10. groups: PEfAPSa 100 mg/kg b.w. and F6Sa 100 mg/kg b.w. should be added, since you presented them in the Figure 3A-C. Please recheck the doses for the acarbose and canagliflozin in the section 2.10. and in the Figure 3A-C, and in the Lines 433 and 435.
Line 183: group (V) not VI
Table 1. Specify what the Table 1 represents! Because how it is now, it looks like that the Table 1 represents body weight instead of blood glucose level (mg/dL).
Line 219: at the end of the sentence add “(Figure 1 -A)”
Line 221-222: remove the brackets for the doses - 50 and 300 mg/kg
Section 3.2. Subchronic assay, I would suggest that in this section at first you put “Organ weight measurement of PEfAPSa, F6Sa and acarbose on STID mice of subchronic assay” as 3.2.1. subsection, and then to continue with the results from Figure 2 A-D.
Line 294: at the end of the sentence add “(Figure 2 -A)”
Line 299: at the end of the sentence add “(Figure 2 -B)”
Line 317: at the end of the sentence add “(Table 2)”
Line 319: Table 2.- Relative organ weight of the subchronic assay PEfAPSa, it is written Table 1
In the Table 2, in the description of the Table 2 and in the Figure 2 and its’ description the dose of acarbose is 100 mg/kg b.w., while in the Material and methods section (2.9. Sub-chronic assay) you stated that group IV received acarbose 50 mg/kg b.w. (line 167). I believe that the mistake was made in the line 167, so please correct that.
Line 334: 3.2.5 should be 3.2.4.1
Line 343: 3.2.6. should be 3.2.4.2
Line 488: remove (Figure 11)
Section 4. -Discussions: Even though this section is improved, when compared to the previous version, it is still difficult to follow and not clear. Authors should be focused on the obtained results and try to explained them in the best possible way in order to fulfill the standards of the Journal.
Author Response
"Please see the attachment."

Reviewer 3 Report
In the revised version of the manuscript by Jesus Iván Solares Pascasio et al. The authors have addressed all my concerns and I have no further comments.
Author Response
Thank you.